# Functional geometry of the cortex encodes dimensions of consciousness

Zirui Huang [1,2] ✉, George A. Mashour[1,2,3,4] & Anthony G. Hudetz[1,2,3]

Consciousness is a multidimensional phenomenon, but key dimensions such as awareness and wakefulness have been described conceptually rather than neurobiologically. We hypothesize that dimensions of consciousness are encoded in multiple neurofunctional dimensions of the brain. We analyze cortical gradients, which are continua of the brain's overarching functional geometry, to characterize these neurofunctional dimensions. We demonstrate that disruptions of human consciousness – due to pharmacological, neuro-pathological, or psychiatric causes – are associated with a degradation of one or more of the major cortical gradients depending on the state. Network-specific reconfigurations within the multidimensional cortical gradient space are associated with behavioral unresponsiveness of various etiologies, and these spatial reconfigurations correlate with a temporal disruption of struc-tured transitions of dynamic brain states. In this work, we therefore provide a unifying neurofunctional framework for multiple dimensions of human con-sciousness in both health and disease.

The nature and mechanism of consciousness are fundamental ques-tions in science, with important implications for multiple medical specialties. Consciousness is a multifaceted phenomenon that can be altered by various causes. For example, normal wakeful consciousness is altered during dreaming, sedation, general anesthesia, post-comatose conditions, or hallucinations, and these states are asso-ciated with both convergent and divergent changes in neural activity.

Consciousness has been defined conventionally by two separable components: awareness of the environment and the self (i.e., content of consciousness), and wakefulness (i.e., level of consciousness)[1]. For example, patients with unresponsive wakefulness syndrome (UWS) can maintain eye-opening and sleep–wake cycles but are presumably unaware of themselves and their surroundings; thus, their condition is classified as wakefulness without awareness. This two-dimensional scheme was recently extended by the inclusion of a behavioral dimension (i.e., ability to produce motor responses)[2], allowing for a better characterization of cognitive-motor dissociation[3] or covert consciousness[4], which occurs in certain behaviorally non-responsive patients with neuropathological disorders. Although multidimensional representations of consciousness have been proposed from a

theoretical perspective[5–8], an important knowledge gap has been the identification of what these dimensions are in neural terms and how they are associated with the brain's neurofunctional properties.

The traditional functional localization approach often fails to link brain regions exclusively to specific functions. For example, the pre-frontal cortex is involved in numerous functions, such as working memory, decision-making, attention, and task-control[9]. At the systems level, a single brain network can also be involved in multiple cognitive processes[10]. Hence, isolated brain regions and networks are unlikely to map onto a single neurofunctional dimension. In the search for the neural correlates of consciousness[11], investigators have typically examined discrete brain regions rather than continua reflecting the brain's intrinsic functional geometry. We hypothesize that dimensions of consciousness are encoded in multiple neurofunctional dimensions of the brain, and a change in a conscious state is fundamentally the manifestation of a change along one or more neurofunctional dimension.

Recent advances in neuroimaging offer a new approach that can be applied to investigate the neurobiology of consciousness. Cortical gradients, which subsume multiple functions and functional networks

[1]Department of Anesthesiology, University of Michigan Medical School, Ann Arbor, MI 48109, USA. [2]Center for Consciousness Science, University of Michigan Medical School, Ann Arbor, MI 48109, USA. [3]Neuroscience Graduate Program, University of Michigan, Ann Arbor, MI 48109, USA. [4]Department of Pharma-cology, University of Michigan Medical School, Ann Arbor, MI 48109, USA. ✉e-mail: huangzu@med.umich.edu

along a continuum from the unimodal systems that underpin perception and action to the association cortices implicated in abstract cognitive functions[12–14], are compelling candidates for the neurofunctional dimensions of consciousness.

Complementing the functional geometry are the brain dynamics that continuously shape and reshape functional configurations, with the spatial patterns of functional connectivity evolving over time. Recent empirical studies suggest that the spatial and temporal properties of brain activity interact through multiple spatiotemporal scales, and that they cannot be understood fully in isolation. In particular, the transient, cortex-wide fMRI co-activations[15–17] propagate as waves traversing along the spatially defined cortical gradients[18–20]. Thus, the temporal dynamics are likely to be shaped by the functional geometry. Understanding the covariation between spatial and temporal factors might thus provide more insight into the neural basis of consciousness than considering them separately.

In this work, we aim to establish a multidimensional neurofunctional account of consciousness. We test three hypotheses as follows (Fig. 1): (1) different neurofunctional dimensions of consciousness are represented by different cortical gradients, and the disruption of consciousness is associated with a degradation of one or more of the major cortical gradients, depending on the state; (2) cortical gradients construct a virtual multidimensional space where depressed or altered states of consciousness are associated with both common and state-specific alterations; (3) reconfigurations of brain network functional geometry are associated with a disruption of structured transitions of dynamic brain states, defined previously as a temporal circuit[15]. We test our hypotheses by comparing cortical gradients among large cohorts of healthy awake and anesthetized participants as well as patients with neuropathological and psychiatric diagnoses. We provide an empirical basis for a unifying neurofunctional framework in which (1)

dimensions of consciousness are represented in neural terms, (2) these dimensions are characteristically altered depending on the state of consciousness, and (3) changes in the brain's functional geometry are associated with shifts in temporal dynamics.

## Results

We analyzed five fMRI datasets from three independent research sites (see Methods for more details[15,21];) including: propofol deep sedation (PDS; drug effect site concentration = ~2.4 μg/ml) in Dataset-1, propofol general anesthesia (PGA; drug effect site concentration = 4.0 μg/ml) in Dataset-2, ketamine anesthesia (KA) in Dataset-3, UWS in Dataset-4, and schizophrenia (SCHZ), bipolar disorder (BD), and attention deficit hyperactivity disorder (ADHD) in Dataset-5. The participants with PDS, PGA, KA, and patients with UWS were behaviorally unresponsive to verbal commands, and they are hereafter referred to as depressed states of consciousness (see Discussion for the distinction between behavioral responsiveness and consciousness). Furthermore, these depressed states of consciousness have both common and unique characteristics with respect to awareness, sensory organization, and arousability. For example, PDS suppresses awareness with preserved arousability to painful glabellum stimulation; PGA induces loss of awareness and loss of arousability[22]; KA is associated with sensory disorganization and partially preserved internal awareness and arousability[23–25]; and UWS is associated with disrupted awareness with preserved arousability and sensory organization[1] (Fig. 2a).

The cortical gradient analysis was used to map the functional brain connectome in a nonlinear diffusion space[12]. The method decomposes the functional similarity structure of the fMRI data into a set of embedding components (i.e., gradients) that describe major spatial axes along which functional connectivity varies at a cortex-wide level. Within each gradient, brain regions are organized by the

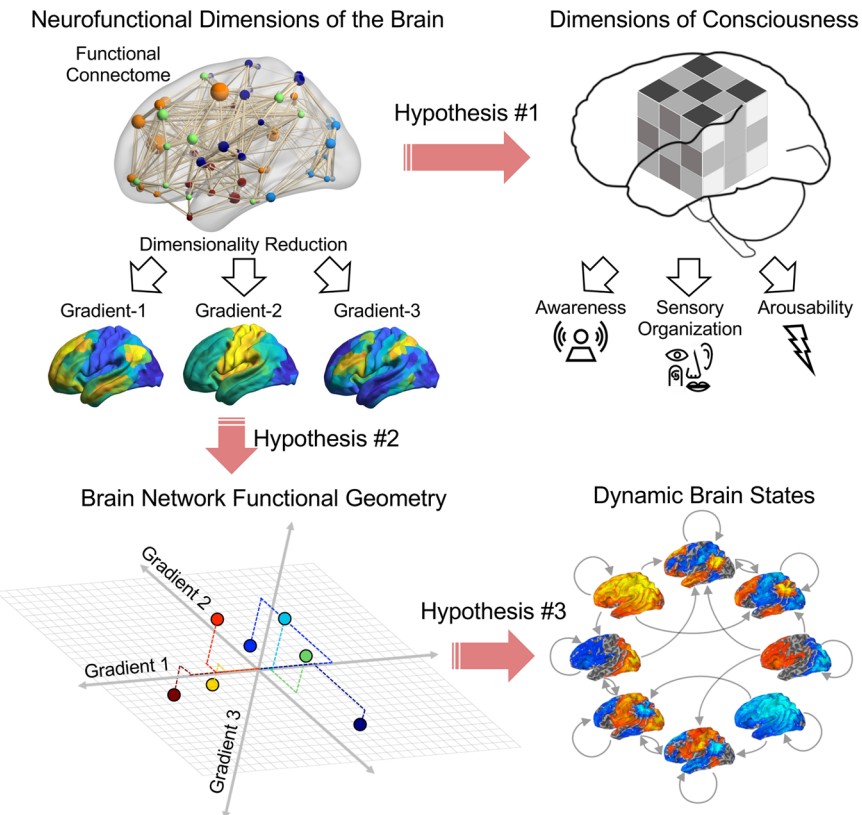

**Fig. 1 | Schematic illustration of three working hypotheses.** Hypothesis-1: dimensions of consciousness are encoded in multiple neurofunctional dimensions of the brain. Hypothesis-2: cortical gradients construct a virtual multidimensional space, in which canonical functional brain networks occupy characteristic positions. Hypothesis-3: brain network functional geometry shapes dynamic brain states.

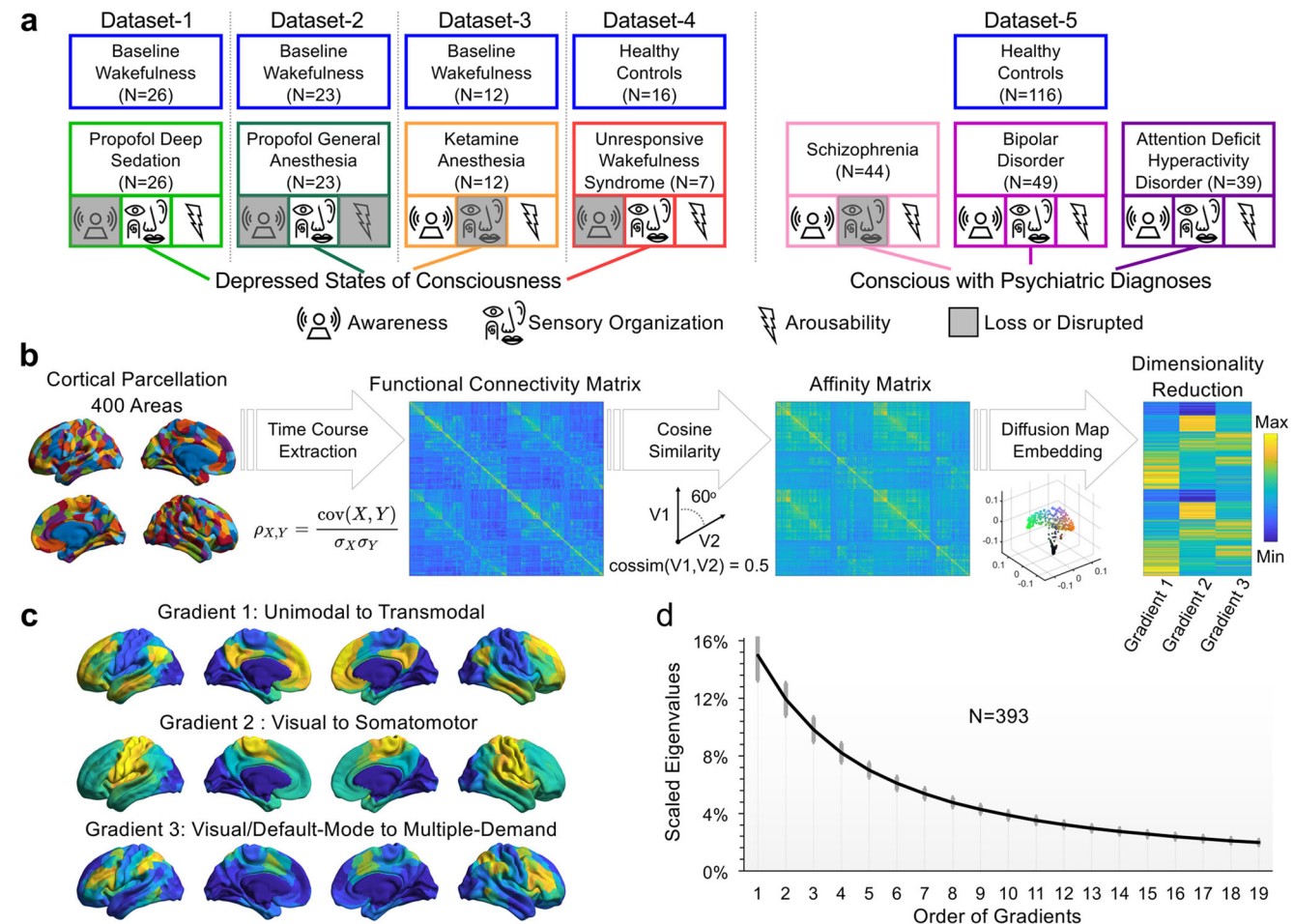

**Fig. 2 | Datasets description and cortical gradient calculation. a** Five fMRI datasets consist of participants and patients with normal wakefulness (baseline/healthy controls) and depressed (or altered) states of consciousness. **b** An overview of cortical gradient analysis. The fMRI time courses were extracted from 400 cortical areas according to a brain parcellation scheme[26]. A 400 × 400 functional connectivity matrix was calculated for each participant and each condition. A normalized cosine angle affinity matrix was calculated to capture the similarity of connectivity profiles between cortical areas. Cortical gradients were computed using a diffusion map embedding algorithm. **c** Gradient-1 ranges from unimodal primary sensory areas to transmodal cortex. Gradient-2 ranges from visual to somatomotor cortices. Gradient-3 ranges from visual/default-mode to areas commonly involved in multiple-demand tasks. **d** Connectome-level variance explained by the gradients (mean ± SD across all scans). The first three gradients explained ~37% of the variance in the functional connectivity matrices. Source data are provided as a Source Data file.

similarity of their observed activity patterns. Brain regions are grouped at one end of a gradient having similar fluctuations in activity over time, and they collectively show less similarity (i.e., more functionally differentiated) to the cluster of regions at the other end of the gradient. Briefly, the fMRI time courses were extracted from 400 cortical areas according to a well-established brain parcellation scheme[26]. A pair-wise functional connectivity matrix was calculated for each participant and each condition, and it was transformed into a normalized cosine angle affinity matrix capturing the similarity of functional connectivity profiles among cortical areas. Cortical gradients were computed using a diffusion map embedding algorithm[12] to identify cortical gradient components (Fig. 2b and Fig. 2c). The first three cortical gradients explained ~37% of the variance in the functional connectivity matrices across the cortex (Fig. 2d). The first axis (Gradient-1) depicts a gradient ranging from unimodal primary sensory areas (visual and somatosensory) to the transmodal cortex, e.g., frontoparietal and default-mode network (mean ± SD of variance explained: 15.0 ± 1.7%). The second axis (Gradient-2) depicts a gradient ranging from visual to somatomotor cortices (11.9 ± 1.1%). The final axis (Gradient-3) depicts a gradient ranging from visual and default-mode areas to the areas commonly implicated in multiple-demand (9.8 ± 0.9%). These gradients and total variance explained (from 30% to 50%) are in agreement with what has been reported previously in the literature[12,27–31].

## Degradation of macroscale cortical gradients

We tested whether a disruption of consciousness is associated with degradation, i.e., less functional differentiation of the gradient extremes, of one or more of the major cortical gradients, depending on the state. We quantified the numerical ranges of each gradient (distance from the minimum to the maximum gradient eigenvector values) and compared them between baseline condition (wakefulness, or healthy control) and depressed states of consciousness or psychiatric diagnoses (Fig. 3a; see Supplementary Fig. 1 for gradients projected onto the cortex for each condition). Statistical analyses were performed separately for each group against their own baseline conditions (or healthy control groups). We used this statistical strategy to account for potential biases due to different experimental and scanning conditions across research sites or datasets. Specifically, Bayesian Paired Samples T-Tests (two-tailed) were performed for PDS, PGA, and KA against their own baseline conditions; Bayesian Independent Samples T-Tests (two-tailed) were performed for UWS, SCHZ, bipolar disorder, and ADHD against their own healthy control groups. For all results, we calculated Bayes factors ($BF_{10}$) and

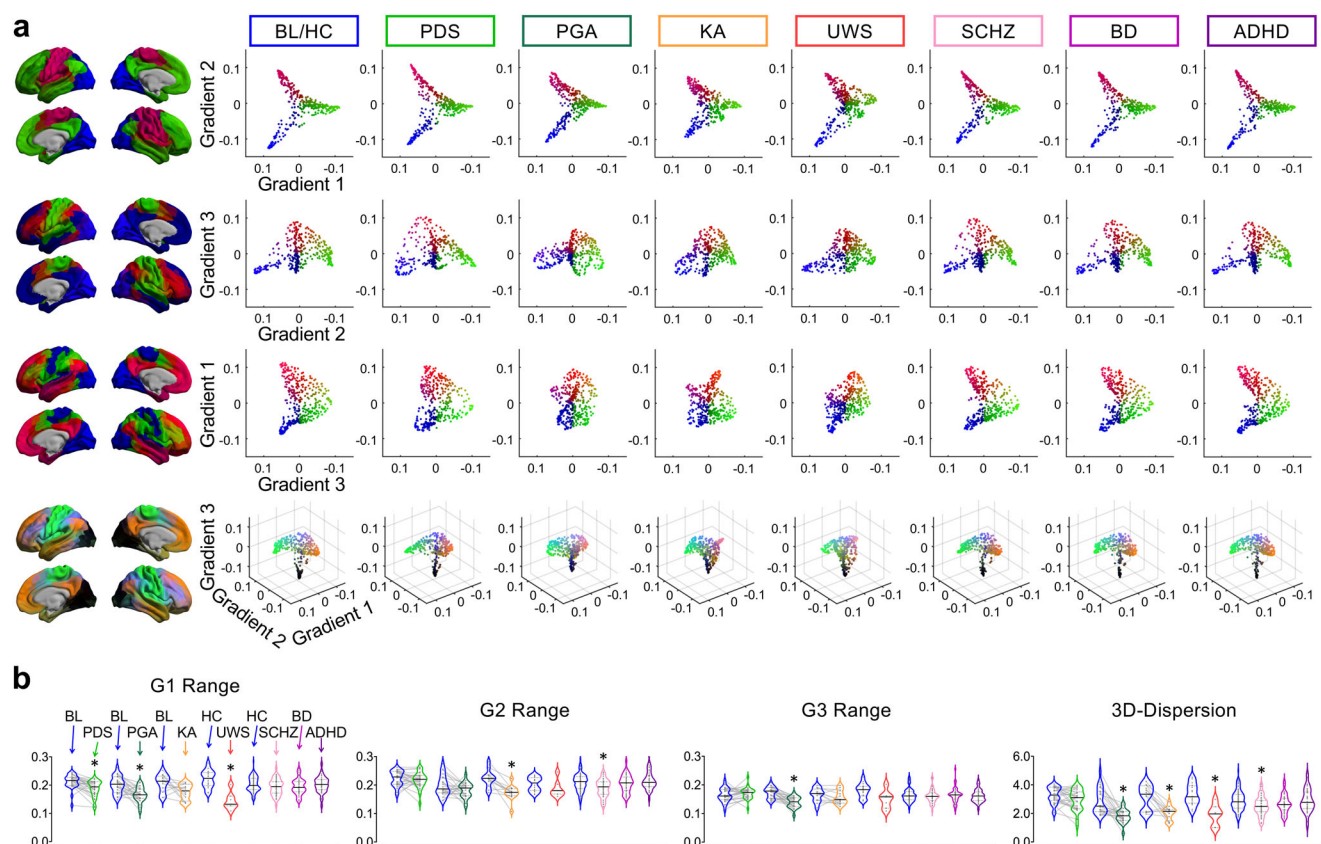

**Fig. 3 | Cortical gradients in various states of consciousness. a** Visualization of the first three cortical gradients in 2D and 3D view. Diffusion map embedding was applied on the group-averaged functional connectivity matrices in conscious baseline and healthy controls (BL/HC; pooled across all datasets), propofol deep sedation (PDS), propofol general anesthesia (PGA), ketamine anesthesia (KA), unresponsive wakefulness syndrome (UWS), schizophrenia (SCHZ), bipolar disorder (BD), and attention deficit hyperactivity disorder (ADHD). **b** Violin plots (median: solid line; quantiles: dash line) of the numerical ranges in each gradient, and 3D-dispersion (sum squared Euclidean distance of all regions to the global

centroid in the 3D cortical gradient space) across all conditions. Bayesian paired samples $T$ tests (two-tailed) were performed for PDS ($n = 26$), PGA ($n = 23$), and KA ($n = 12$) against their own baseline conditions ($n = 26$, $n = 23$, $n = 12$, respectively). Bayesian Independent Samples T-Tests (two-tailed) were performed for UWS ($n = 7$), SCHZ ($n = 44$), BD ($n = 49$), and ADHD ($n = 39$) against their own healthy control groups ($n = 16$, $n = 116$, $n = 116$, $n = 116$, respectively). Asterisk indicates $BF_{10} > 10$. Detailed statistics are provided in Supplementary Data 1. Source data are provided as a Source Data file.

reported the posterior median with 95% credible interval (CI). We applied the default effect size priors, Cauchy scale 0.707, and used Jeffreys' criterion for the interpretation of the strength of evidence[32]. Due to the numerous statistical tests performed on different measures across datasets, we sought to minimize potential false inferences by excluding anecdotal ($BF_{10} = 1–3$) and moderate ($BF_{10} = 3–10$) evidence, while collecting strong ($BF_{10} = 10–30$), very strong ($BF_{10} = 30–100$), and decisive ($BF_{10} = 100$) evidence for ($H_1$: there is effect) or against ($H_0$: there is no effect) either hypothesis[32]. Regardless, the strength of the evidence can be judged directly from the numerical value of $BF_{10}$ reported in Supplementary Data 1. $T$ and $p$ values (false discovery rate-corrected at $\alpha = 0.05$) derived from Classical Student's $T$ tests were also reported to complement Bayesian statistics.

For the Gradient-1 range, we found decisive evidence for a reduction in PDS ($t_{(25)} = 4.331$, $p = 0.002$, $BF_{10} = 136.507$, 0.785 [CI: 0.344, 1.240]) and UWS ($t_{(21)} = 5.660$, $p < 0.001$, $BF_{10} = 1098.829$, 2.301 [CI: 1.079, 3.536]), and very strong evidence for a reduction in PGA ($t_{(22)} = 4.000$, $p = 0.002$, $BF_{10} = 55.220$, 0.763 [CI: 0.229, 1.244]). For the Gradient-2 range, we found decisive evidence for a reduction in KA ($t_{(11)} = 5.170$, $p = 0.001$, $BF_{10} = 109.209$, 1.312 [CI: 0.502, 2.189]) and SCHZ ($t_{(158)} = 3.772$, $p = 0.006$, $BF_{10} = 102.473$, 0.624 [CI: 0.275, 0.979]). For the Gradient-3 range, we found decisive evidence for a reduction in PGA ($t_{(22)} = 5.494$, $p < 0.001$, $BF_{10} = 1416.786$, 1.063 [CI: 0.541, 1.605]) (Fig. 3b; Supplementary Data 1).

We further quantified the global dispersion, i.e., sum squared Euclidean distance of all regions to the global centroid, in the three-dimensional cortical gradient space. A small dispersion value indicates that the overall functional connectivity profiles across regions have low differentiation. Accordingly, we found decisive evidence for a reduction in PGA ($t_{(22)} = 4.736$, $p < 0.001$, $BF_{10} = 272.220$, 0.910 [CI: 0.418, 1.420]), very strong evidence for a reduction in UWS ($t_{(21)} = 4.182$, $p = 0.001$, $BF_{10} = 58.409$, 1.614 [CI: 0.533, 2.728]), and strong evidence for a reduction in KA ($t_{(11)} = 3.781$, $p = 0.006$, $BF_{10} = 15.735$, 0.938 [CI: 0.253, 1.685]) and SCHZ ($t_{(158)} = 3.076$, $p = 0.013$, $BF_{10} = 12.948$, 0.504 [CI: 0.162, 0.854]) (Fig. 3b; Supplementary Data 1).

Together, these results suggest that a specific combination of gradient changes identify each depressed or altered state of consciousness. That is, PDS, PGA, and UWS were associated with a degradation of gradient between unimodal and transmodal areas (Gradient-1); KA and SCHZ were associated with a degradation of gradient between visual and somatomotor areas (Gradient-2); and high-dose propofol (i.e., PGA) further collapsed the functional gradient between visual/default-mode to multiple-demand areas (Gradient-3).

## Common and specific network alterations

We hypothesized that the foregoing three cortical gradients construct a virtual 3D gradient space, where depressed or altered states of consciousness are associated with both common and state-specific

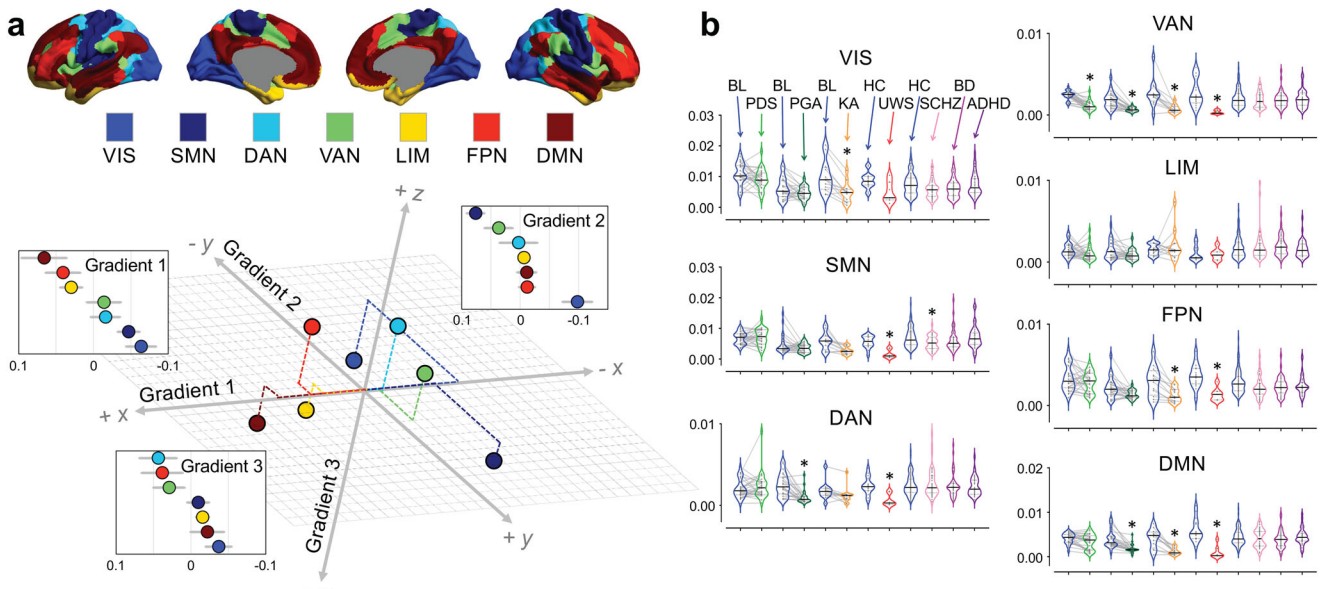

**Fig. 4 | Network eccentricity in various states of consciousness. a** Seven pre-defined functional networks include the visual network (VIS), somatomotor network (SMN), dorsal attention network (DAN), ventral attention/salience network (VAN), limbic network (LIM), frontoparietal network (FPN), and default-mode network (DMN). Schematic illustration of network position in a 3D gradient space. Diffusion map embedding was applied to the group-averaged functional connectivity matrix in conscious baseline and healthy controls (*n* = 193; pooled across all datasets). Gradient eigenvector loading values (mean ± SD across areas within a given network) are shown for each network. Dashed lines indicated the relative distance from global centroid to each network's centroid along each axis. **b** Violin plots (median: solid line; quantiles: dash line) of network eccentricity, measured by

the Euclidean distance of a given network's centroid to the global centroid, across all conditions. Bayesian Paired Samples *T* tests (two-tailed) were performed for propofol deep sedation (PDS; *n* = 26), propofol general anesthesia (PGA; *n* = 23), and ketamine anesthesia (KA; *n* = 12) against their own baseline conditions (BL; *n* = 26, *n* = 23, *n* = 12, respectively). Bayesian Independent Samples *T* tests (two-tailed) were performed for unresponsive wakefulness syndrome (UWS; *n* = 7), schizophrenia (SCHZ; *n* = 44), bipolar disorder (BD; *n* = 49), and attention deficit hyperactivity disorder (ADHD; *n* = 39) against their own healthy control groups (HC; *n* = 16, *n* = 116, *n* = 116, *n* = 116, respectively). Asterisk indicates BF$_{10}$ > 10. Detailed statistics are provided in Supplementary Data 1. Source data are provided as a Source Data file.

alterations. Accordingly, canonical functional brain networks occupy characteristic positions in this space, and the network-level functional geometry (e.g., distance) can be quantified. We measured network eccentricity (Euclidean distance between a given network's centroid and the global centroid), and pair-wise network distance (Euclidean distance between the centroids of two networks) in the 3D gradient space (Fig. 4a). These multidimensional distance metrics reflect the similarity of connectivity profiles across networks based on multiple axes of differentiation[27]. The network analyses were performed for seven pre-defined functional networks including the default-mode network (DMN), lateral frontoparietal network (FPN), limbic network (LIM), ventral attention/salience network (VAN), dorsal attention network (DAN), somatomotor network (SMN), and visual network (VIS)[26].

For network eccentricity during PDS, we found decisive evidence for a reduction in VAN ($t_{(25)}$ = 8.677, $p$ < 0.001, BF$_{10}$ = 2.393e + 6, 1.617 [CI: 1.018, 2.242]). For PGA, we found decisive evidence for a reduction in DAN ($t_{(22)}$ = 5.009, $p$ < 0.001, BF$_{10}$ = 493.899, 0.965 [CI: 0.462, 1.487]), VAN ($t_{(22)}$ = 6.237, $p$ < 0.001, BF$_{10}$ = 6940.439, 1.214 [CI: 0.661, 1.789]) and DMN ($t_{(22)}$ = 4.957, $p$ < 0.001, BF$_{10}$ = 440.966, 0.995 [CI: 0.454, 1.474]; parametric test assumption was violated; non-parametric test: $W$ = 270, $p$ < 0.001, BF$_{10}$ = 1431.364, 1.380 [CI: 0.663, 2.037]). For KA, we found decisive evidence for a reduction in VIS ($t_{(11)}$ = 5.224, $p$ = 0.001, BF$_{10}$ = 117.198, 1.326 [CI: 0.512, 2.209]), and very strong evidence for a reduction in VAN ($t_{(11)}$ = 4.387, $p$ = 0.003, BF$_{10}$ = 37.348, 1.100 [CI: 0.362, 1.902]; parametric test assumption was violated; non-parametric test: $W$ = 77, $p$ = 0.003, BF$_{10}$ = 170.249, 1.490 [CI: 0.535, 2.690]), FPN ($t_{(11)}$ = 4.281, $p$ = 0.003, BF$_{10}$ = 32.192, 1.072 [CI: 0.343, 1.864]), and DMN ($t_{(11)}$ = 4.984, $p$ = 0.001, BF$_{10}$ = 85.025, 1.261 [CI: 0.469, 2.120]). For UWS, we found decisive evidence for a reduction in SMN ($t_{(21)}$ = 4.871, $p$ < 0.001, BF$_{10}$ = 226.944, 1.935 [CI: 0.783, 3.103]) and DMN ($t_{(21)}$ = 4.768, $p$ < 0.001, BF$_{10}$ = 185.148, 1.888 [CI:

0.745, 3.047]), very strong evidence for a reduction in DAN ($t_{(21)}$ = 3.960, $p$ = 0.002, BF$_{10}$ = 38.120, 1.511 [CI: 0.456, 2.607]) and FPN ($t_{(21)}$ = 3.989, $p$ = 0.002, BF$_{10}$ = 40.277, 1.524 [CI: 0.466, 2.623]), and strong evidence for a reduction in VAN ($t_{(21)}$ = 3.716, $p$ = 0.002, BF$_{10}$ = 24.030, 1.397 [CI: 0.373, 2.476]; parametric test assumption was violated; non-parametric test: $W$ = 110, $p$ = 0.002, BF$_{10}$ = 8.307, 1.071 [CI: 0.168, 2.066]). For SCHZ, we found strong evidence for a reduction in SMN ($t_{(158)}$ = 3.254, $p$ = 0.009, BF$_{10}$ = 21.188, 0.535 [CI: 0.190, 0.886]; parametric test assumption was violated; non-parametric test: $W$ = 3326, $p$ = 0.017, BF$_{10}$ = 7.061, 0.457 [CI: 0.122, 0.810]). No strong evidence was found in either bipolar disorder or ADHD (Fig. 4b; Supplementary Data 1). Together, we identified both common and specific changes in network eccentricity in various states of consciousness.

For pair-wise network distance, the main results are summarized as follows (Fig. 5; see Supplementary Data 1 for detailed statistics): (1) shortened network distance of VAN-DMN appeared to be a unitary correlate of depressed states of consciousness, albeit moderate evidence (BF$_{10}$ = 6.833) was found in UWS for non-parametric test; (2) shortened network distance of DAN-VIS, DMN-SMN (non-parametric test for PGA), DMN-DAN, FPN-VAN, and DMN-FPN was seen in PGA, KA, and UWS, but not in PDS; (3) Shortened network distance of SMN-VIS was shared by both KA and SCHZ. However, KA and SCHZ differed in either SMN or VIS, which dominates the network changes. Specifically, KA was associated with VIS-dominant changes including the shortened network distance of SMN-VIS, DAN-VIS, VAN-VIS, FPN-VIS, and DMN-VIS. In contrast, SCHZ was associated with SMN-dominant changes including the shortened network distance of SMN-VIS, VAN-SMN, and FPN-SMN; (4) Bipolar disorder was associated with shortened network distance of DMN-VIS. No strong evidence for shortened network distance was found in ADHD.

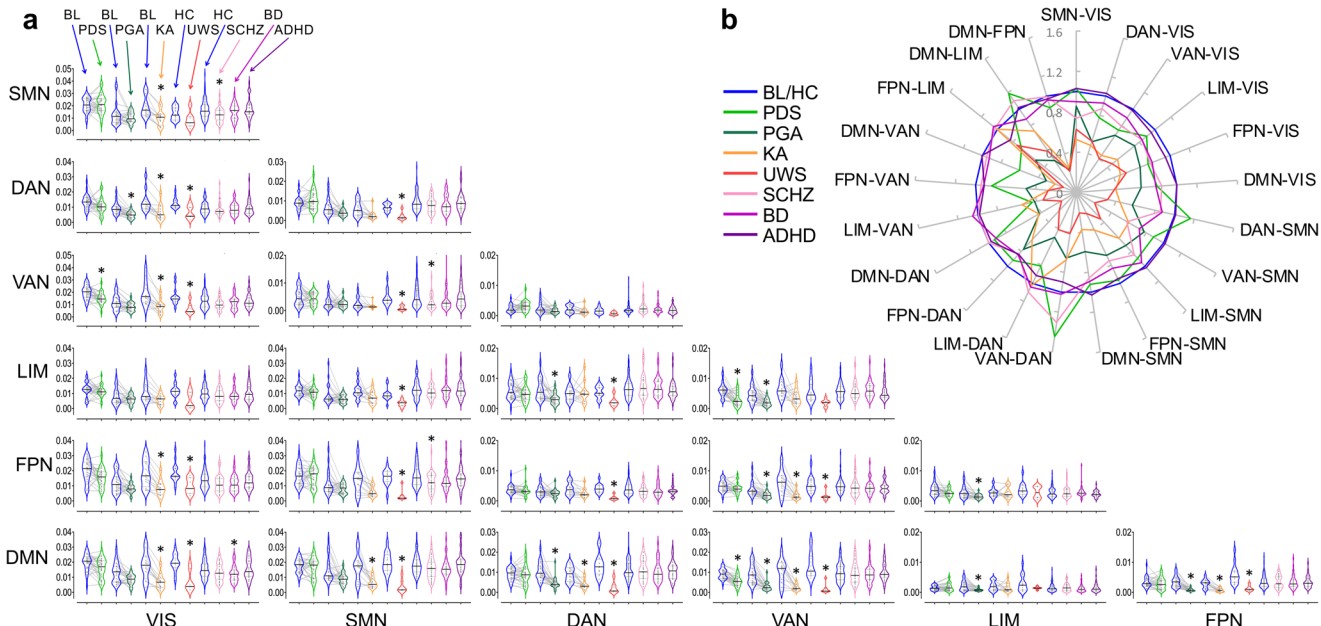

**Fig. 5 | Network distance in various states of consciousness. a** Violin plots (median: solid line; quantiles: dash line) of pair-wise network distance, measured by the Euclidean distance between the centroids of two networks in the 3D gradient space, across all conditions. Bayesian paired samples *T* tests (two-tailed) were performed for propofol deep sedation (PDS; $n = 26$), propofol general anesthesia (PGA; $n = 23$), and ketamine anesthesia (KA; $n = 12$) against their own baseline conditions (BL; $n = 26$, $n = 23$, $n = 12$, respectively). Bayesian Independent Samples *T* tests (two-tailed) were performed for unresponsive wakefulness syndrome (UWS; $n = 7$), schizophrenia (SCHZ; $n = 44$), bipolar disorder (BD; $n = 49$), and attention deficit hyperactivity disorder (ADHD; $n = 39$) against their own healthy control groups (HC; $n = 16$, $n = 116$, $n = 116$, $n = 116$, respectively). Asterisk indicates $BF_{10} > 10$. **b** Spider plots displaying mean values of network distance across all conditions. Values for each depressed state of consciousness or psychiatric diagnosis were normalized by dividing the mean of their own baseline (or healthy control) conditions. VIS visual network, SMN somatomotor network DAN dorsal attention network, VAN ventral attention/salience network, LIM limbic network, FPN frontoparietal network, DMN default-mode network. Detailed statistics are provided in Supplementary Data 1. Source data are provided as a Source Data file.

## Cortical gradients are linked to dynamic brain activity

Next, we tested whether the network relationship (e.g., functional distance) in 3D gradient space could inform how brain activity evolves over time. We focused on a three-way network relation, i.e., VAN-DAN-DMN[33,34], which has been shown to be particularly relevant to conscious processing[15,21]. Adopting co-activation analysis[15], we calculated the occurrence rates of the co-activation patterns (i.e., dividing the number of fMRI volumes belonging to a given co-activation pattern by the total number of volumes per scan) consisting of VAN, DAN, and DMN; hereafter referred to as VAN+, DAN+, and DMN+. The co-activation patterns were identified by an unsupervised machine-learning approach (i.e., k-means clustering algorithm) that assessed areas across the brain that are consistently activated together rather than averaging activity over long periods (Fig. 6).

We uncovered a tight link between the network distance of VAN, DAN, and DMN, and the temporal expression property (i.e., occurrence rate) of VAN+, DAN+, and DMN+, based on Bayesian Kendall Correlations. The following results were reported with Kendall's tau-b (τb), FDR-corrected *p* values (at $\alpha = 0.05$), $BF_{10}$, and 95% CI. We found decisive evidence for a negative correlation between DMN-VAN distance and the occurrence rates of VAN+ (τb = −0.354, $p < 0.001$, $BF_{10} = 4.288e + 22$, CI: −0.418, −0.286), and decisive evidence for a positive correlation between DMN-VAN distance and the occurrence rates of DAN+ (τb = 0.459, $p < 0.001$, $BF_{10} = 7.248e + 38$, CI: 0.390, 0.521) and DMN+ (τb = 0.476, $p < 0.001$, $BF_{10} = 6.320e + 41$, CI: 0.406, 0.538). Similar correlation profiles were found between DMN-DAN distance and the occurrence rates of VAN+ (τb = −0.335, $p < 0.001$, $BF_{10} = 1.402e + 20$, CI: −0.399, −0.267), DAN+ (τb = 0.454, $p < 0.001$, $BF_{10} = 8.169e + 37$, CI: 0.384, 0.516), and DMN+ (τb = 0.502, $p < 0.001$, $BF_{10} = 5.003e + 46$, CI: 0.432, 0.564). However, the correlations between VAN-DAN distance and the occurrence rates of VAN+ (τb = 0.052, $p = 0.151$, $BF_{10} = 0.215$, CI: −0.015, 0.117), DAN + (τb =

0.048, $p = 0.168$, $BF_{10} = 0.178$, CI: −0.019, 0.113) and DMN+ (τb = 0.085, $p = 0.016$, $BF_{10} = 1.554$, CI: 0.019, 0.150) were much weaker. Together, a reduced functional differentiation of DMN-VAN and DMN-DAN (as seen in the depressed states of consciousness) was associated with decreased occurrence rates of DMN+ and DAN+ but with an increased occurrence rate of VAN+.

## Control analyses

Control analyses were performed to test the robustness of results. (1) In line with previous studies using cortical gradient analysis[29,35], our findings were unaffected by the optional use of global signal regression (GSR) during data preprocessing (Supplementary Fig. 2, Supplementary Fig. 3, Supplementary Data 2). (2) The fMRI scanning time per state was different across datasets ranging from 22 min in Dataset-1 to 5 min in Dataset-5. To account for a potential confound due to different scanning times, we trimmed the data length with a fixed duration of 5 min starting from the onset of each scan, then re-evaluated the key results including the gradient ranges and network distance of VAN-DMN. The results agreed with our original findings (Supplementary Fig. 4, Supplementary Data 3). (3) In our original analysis of cortical gradients, we thresholded the connectivity matrix at the sparsity of 90% (i.e., leaving only top 10% of weighted connections per row) and set parameter $\alpha$ at 0.5 ($\alpha$ controls the influence of the density of sampling points). Although both settings were in accordance with previous recommendations[12,27–31], we explored if they were indeed optimal for our analysis. Therefore, we reanalyzed the data by varying the sparsity from 0% to 90% (by 10% increments) and varying the parameter $\alpha$ from 0 to 1 (by 0.1 increments). We found that different sparsity settings impact the group comparison results of gradient range, whereas they showed negligible impact on the results of VAN-DMN network distance (Supplementary Fig. 4, Supplementary Data 3). For example, the difference between SCHZ and healthy controls in

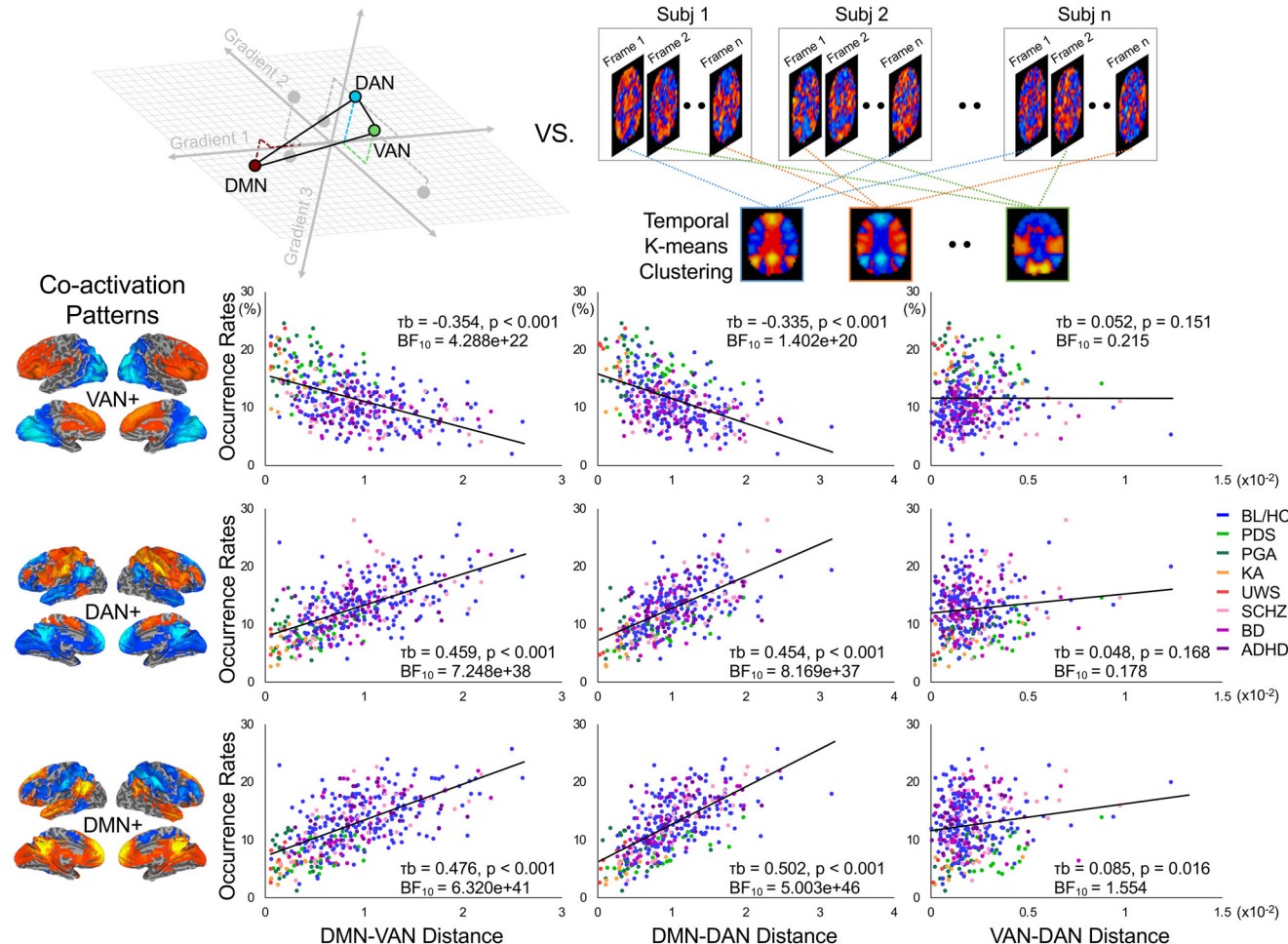

**Fig. 6 | Correlation between-network distance and occurrence rates of co-activation patterns.** Pair-wise network distance was calculated among ventral attention/salience network (VAN), dorsal attention network (DAN), and default-mode network (DMN). Co-activation analysis was applied to calculate their occurrence rates, namely VAN+, DAN+, and DMN+. Bayesian Kendall Correlations were performed between-network distance (DMN-VAN, DMN-DAN, and VAN-DAN) and occurrence rates (VAN+, DAN+, and DMN+) across all participants and conditions ($n = 393$). Kendall's tau-b ($\tau b$), FDR-corrected $p$ values (at $\alpha = 0.05$), and $BF_{10}$ are reported in each scatter plot. A smaller network distance of DMN-VAN and DMN-DAN was associated with a suppression of occurrence in DMN+ and DAN+, with an increased probability of the occurrence in VAN+. BL/HC conscious baseline and healthy controls, PDS propofol deep sedation, PGA propofol general anesthesia, KA ketamine anesthesia, UWS unresponsive wakefulness syndrome, SCHZ schizophrenia, BD bipolar disorder, ADHD attention deficit hyperactivity disorder. Source data are provided as a Source Data file.

Gradient-2 range was only detected at the sparsity of 90%. This may suggest that sparsity with 90% was an optimal choice, but it warrants more methodological investigations that are beyond the scope of our study. Finally, all results were robust with respect to the choice of parameter $\alpha$ (Supplementary Fig. 4, Supplementary Data 3).

## Discussion

Analyzing functional neuroimaging data obtained from healthy participants as well as patients with neuropathological and psychiatric conditions, we provide evidence for three candidate cortical gradients that reflect key neurofunctional dimensions of consciousness, i.e., awareness, sensory organization, and arousability. Gradient-1 was degraded in deep sedation or general anesthesia with propofol and in patients with UWS, indicating loss of awareness. Gradient-1 was less degraded during KA, consistent with well-established reports of disconnected consciousness such as dream-like experiences. Instead, KA produced a notable degradation of Gradient-2, suggesting sensory disorganization. Gradient-3 degraded when participants were no longer arousable by painful glabellum stimulation during propofol infusion at a dose sufficient for general anesthesia. Furthermore, reduced network distance between VAN and DMN in the 3D cortical

gradient space was found to correlate with behavioral unresponsiveness. Relative to normal wakefulness or healthy controls, KA and SCHZ showed some distinct network changes, but both were associated with reduced functional differentiation between VIS and SMN. Finally, a three-way network functional geometry of VAN-DMN-DAN and the temporal occurrence rates of their corresponding co-activation patterns covaried with the state of consciousness. Together, these results provide an empirical basis for a unifying neurofunctional account of consciousness.

Human consciousness is supported by integrated brain activity across functionally segregated areas[36]. A delicate balance between functional integration and differentiation across the cortex seems to be a prerequisite for the coexistence of several functional gradients[12,13,29]. Here we show that unique neurofunctional dimensions of consciousness can be represented by cortical gradients, and that a specific combination of degradation of these gradients characterizes each depressed or altered state of consciousness (Fig. 7).

The unimodal to transmodal gradient (Gradient-1) represents a functional spectrum ranging from direct perception and action to integration and abstraction of information, which is thought to guide the flow of information across the cortex, allowing sensory signals to

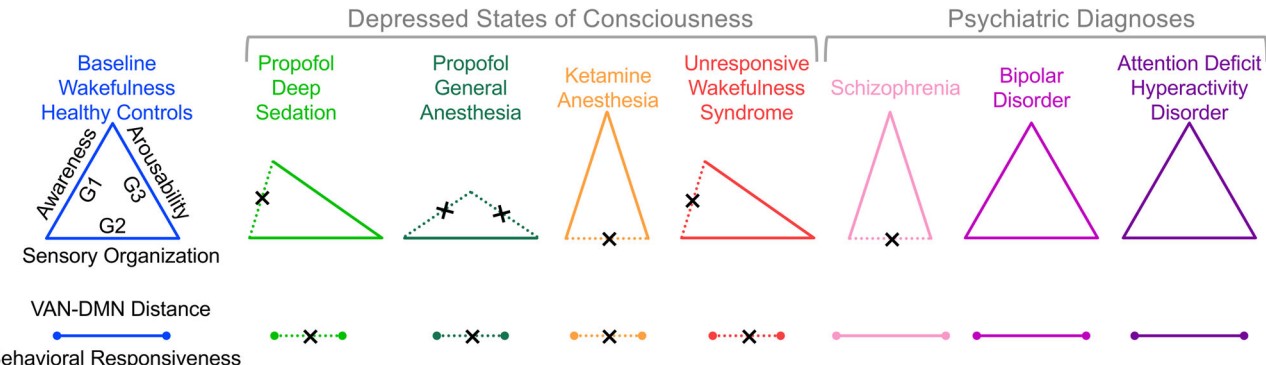

**Fig. 7 | Schematic illustration of the common and specific gradient features of various states of consciousness.** The three cortical gradients (G1, G2, and G3) putatively represent three dimensions of consciousness, i.e., awareness, sensory organization, and arousability. Reduced network distance between ventral attention network (VAN) and default-mode network (DMN) in the 3D gradient space is associated with behavioral unresponsiveness. Dashed lines indicate the major changes during depressed states of consciousness or psychiatric diagnoses as compared to baseline conditions or healthy controls.

become increasingly bound to non-sensory information and transformed into more abstract contents[12–14]. A degradation of this gradient characterized PDS, PGA, and UWS, but not ketamine exposure or psychiatric disorders (SCHZ, BD, and ADHD). On the one hand, this suggests that depressed states of consciousness with an apparent loss of awareness (PDS, PGA, and UWS) are related to a general reduction of functional differentiation between unimodal and transmodal cortices. On the other hand, such a reduction of gradients was not prominent after administration of ketamine. This is consistent with preservation of disconnected awareness during ketamine sedation or anesthesia, which indeed has been previously reported as part of the "dissociative" state induced by this drug[25,37,38]. Furthermore, conscious participants with psychiatric disorders did not show degradation in Gradient-1, which also suggests that the preservation of an intact range along the gradient from low-level sensory processing to abstract cognition is vital for awareness.

The visual-to-somatomotor gradient (Gradient-2) represents a functional specialization of different sensory modalities[12–14]. Consistently, both ketamine and SCHZ were associated with a degradation of this gradient. Prior studies have suggested shared molecular pathways, e.g., dysregulation of N-methyl-d-aspartate receptors[39], and shared phenomenology between ketamine's dissociative/psychoactive effects and schizophrenic symptoms[40,41]. In our study, all participants receiving ketamine reported having dreams, and 8 of 12 participants could recall their hallucinations or reported out-of-body experiences (e.g., flying on a cloud, weird smells, taking an elevator, and thick mist) after recovery. Therefore, we speculate that degradation of functional differentiation between visual and somatomotor gradient may correspond to increased crosstalk between visual and somatosensory information resulting in an altered sensory organization arguably associated with psychoactive effects[24].

The visual/default-mode to multiple-demand gradient (Gradient-3) represents a functional differentiation extending from areas in the visual system and DMN to regions in frontoparietal and attention networks. We showed that degradation of this gradient occurred when participants were no longer arousable by painful glabellum stimulation during a high dose of propofol. We postulate that Gradient-3 is associated with arousal. The conventional concept of arousal relates to the ascending reticular activating system (ARAS), which includes the brainstem, medial thalamus, and basal forebrain. Hence, Gradient-3 is best interpreted as cortical arousal, which could be a result of ARAS activation. Further study is required to identify a covariation between Gradient-3 range and regional activation along the ARAS. Nevertheless, the neurofunctional role of Gradient-3 remains to be determined. First, this gradient was previously identified as paralimbic to multiple-demand[28] or task-negative to task-positive[42] functional axis, and was suggested to be relevant for cognitive processes involving vigilance and working memory[28], and the effect of task demands (off-task vs. on-task) on cognition[42]. Second, a perturbational approach may be useful to ascertain if Gradient-3 is causally related to cortical arousability. Recent work applied dynamical models of whole-brain activity and dissociated the level and stability (or reversibility) of consciousness[43]. Stability refers to the likelihood of a state transitioning from unconsciousness to consciousness after external stimulation. In this respect, it would be of interest to examine if Gradient-3 is related to the reversibility of unconsciousness in response to perturbations.

Based on this evidence, these three cortical gradients are compelling candidates for three neurofunctional dimensions of consciousness, i.e., awareness, sensory organization, and cortical arousability (Fig. 7). This interpretation supports a multidimensional definition of consciousness, where global states (as distinct from local, content-involving states) of consciousness can be conceived of as regions within a multidimensional space[5]. Notably, our work represents a conceptual departure from cognitive-behavioral definitions of consciousness, such as a traditional two-dimensional representation (awareness vs. wakefulness)[1], or multiple cognitive representations such as attention, memory, sensory processing, executive function, and meta-awareness[5–8]. We propose that a given state of consciousness can be represented, independently of cognitive-behavioral description, by a family of cortical gradients derived from the principled characterization of the brain's functional geometry. Instead of focusing on the presumed role of particular brain regions or networks, this proposed framework may obviate the need to debate if neural correlates of consciousness are, for example, in the front or back of the brain[44].

By characterizing network relationships in a 3D gradient space, we found that VAN and DMN are the most consistently vulnerable networks when consciousness is depressed. In particular, a reduced functional differentiation (shortened network distance) between VAN and DMN was associated with all studied depressed states of consciousness (albeit moderate evidence was found in UWS). Although behavioral unresponsiveness is the most pragmatic and commonly used surrogate measure of depressed states of consciousness in both clinical and investigative settings, we acknowledge that behavior can, in certain cases, be dissociated from awareness such as in covert consciousness in patients with neuropathological disorders[4], or ketamine-induced dissociative states[25,37,38]. Hence, behavioral unresponsiveness neither has the causal power for unconsciousness, nor is a reliable indicator of unconsciousness[45]. Accordingly, we argue that behavioral unresponsiveness may not simply result from degradation of single neurofunctional dimension but from multidimensional alterations converging on certain network-specific, i.e., VAN and DMN, reconfigurations.

There is compelling evidence suggesting that the VAN, including the anterior insular cortex, mediates dynamic brain network transitions[33,34] and gates conscious access to sensory information[21]. Therefore, the VAN seems to be an interface between unimodal and transmodal processing, where salient information is identified from unimodal areas, prioritized, and then transferred to transmodal areas (e.g., DMN) for more abstract representations. We speculate that a compression of the functional geometry of VAN-DMN may result in a breakdown of hierarchical processing from sensory signals to abstract contents, leading to a failure of interaction between the agent and the environment.

The stream of consciousness relies on the temporal dynamics of ongoing brain activity and its coordination over distant cortical regions[15]. Recent work showed that transient, cortex-wide fMRI co-activations propagate as waves traversing along the macroscale cortical gradients[18–20]. Canonical functional brain networks have been suggested to be derived from these co-activation patterns, which anchor characteristic positions along the cortical gradients.

In the present study, we showed that the brain's functional geometry (e.g., network distance in the gradient space) and temporal dynamics (e.g., occurrence rates of co-activation patterns) covary with the state of consciousness. The co-activation patterns of DMN+, DAN+, and VAN+, extracted from single fMRI volumes at the temporal resolution of 1–2 s, can be considered snapshots of the aforementioned propagation waves. Our results show that during depressed states of consciousness, the propagation of co-activations across the DMN and DAN (i.e., DMN+ and DAN+) becomes less frequent, whereas those traversing the VAN (i.e., VAN+) become more frequent. These effects are associated with a reduced functional distance of DMN-VAN and DMN-DAN in the 3D gradient space. We suggest that a shift of co-activation occurrence from DMN+ and DAN+ to VAN+ is akin to "traffic congestion" along the cortical gradients, where co-activation propagations are confined to the VAN. Such an interpretation is consistent with the putative role of VAN, which may have the ability to initiate a phase-resetting process that flexibly reroutes a propagation wave of activity between unimodal and transmodal cortices[19,46]. If so, the phase-resetting process should include the antagonistic propagation waves traversing DMN and DAN in the normal conscious state. During depressed states of consciousness, however, dysfunction of VAN may impair propagation rerouting, resulting in reduced occurrences of DMN+ and DAN+. The reduced network distance of DMN-VAN and DMN-DAN may also be accounted for by the disruption of temporal circuits in depressed states of consciousness—an isolation of DMN+ and DAT+ from the trajectory space, which is instead monopolized by giant attractors including the VAT+[15]. Together, these findings advance our understanding of the relationship between the spatial and temporal dimensions of consciousness, specifically, how a change in the brain's functional geometry would affect the brain's temporal dynamics.

Our work may facilitate better understanding of both common and unique neural substrates of the sedative-hypnotic effects of various anesthetics with distinct molecular targets. Propofol is a widely used anesthetic in modern clinical medicine to suppress patient awareness. The primary mechanism of propofol's action is potentiating inhibitory GABA-A receptors, which causes widespread shifts in neuronal activity[36]. At a sufficiently high dose, propofol induces a behavioral state similar to deep coma. The hypnotic effects of ketamine appear to be largely mediated by blockade of N-methyl-d-aspartate receptors[25] and hyperpolarization-activated cyclic nucleotide-gated cation type 1 channels[47]. In contrast to propofol, ketamine induces an atypical state known as "dissociative anesthesia" in which a person is unresponsive to sensory or physical stimuli, but can often experience dream-like states and immersive visions[25,37,38]. We demonstrated that the effects of these two drugs on cortical gradients were dissociable along two different dimensions, i.e., unimodal to

transmodal (Gradient-1) and visual to somatomotor (Gradient-2). A plausible inference is that propofol suppresses awareness by collapsing a unimodal-transmodal hierarchical processing without changing sensory organization, whereas ketamine partially preserves inner subjective experience but distorts sensory experience by collapsing different sensory modalities. In addition, we observed dose-dependent effects of propofol on cortical gradients, where a high dose further collapsed the cortical gradient of visual/default-mode to multiple-demand areas (Gradient-3). Accordingly, there remains a possibility that KA would also show dose-dependent effects on the cortical gradients. For example, a higher dose of ketamine might further collapse Gradient-1 or Gradient-3. Future studies linking the underlying molecular pathways and cortical gradients with different drug doses may be of relevance in testing these hypotheses.

Beyond the field of anesthesiology, identifying the neural correlates of consciousness is important for the diagnosis, treatment, prevention, and rehabilitation for disorders of consciousness in neurology. Multiple neuroimaging studies have demonstrated that covert consciousness could be present in a portion of behaviorally non-responsive patients[4,48–50]. However, there has been controversy over the appropriate methods to detect such covert consciousness and residual cognitive function, which has enormous clinical and ethical interpretations[51,52]. Assuming a lack of response, we are currently unable to determine if a patient or person is: (1) completely unconscious, (2) conscious (e.g., dream state) but disconnected from the environment, (3) conscious and connected to the environment but unable to respond to command, or (4) conscious and connected to the environment but unwilling or unmotivated to respond to command. Considering the above scenarios, it is therefore important to develop behavior-independent and task-independent approaches to assessment. Accordingly, cortical gradient measurements have the potential to reduce the uncertainty of clinical assessment of consciousness in patients who cannot communicate but this requires validation. For example, a degradation of the cortical gradient from unimodal to transmodal areas (Gradient-1) may serve as an indicator of disrupted awareness. Furthermore, we found common gradient features in participants with PDS and patients with UWS, i.e., a degradation of Gradient-1 with relatively preserved Gradient-2 and Gradient-3. The results are consistent with the predictions that both PDS and UWS are associated with disrupted awareness but preserved arousability and sensory organization. Unlike the heterogeneous clinical conditions often associated with UWS, PDS may serve as a pharmacological model in well-controlled experimental settings. For example, machine-learning models can be trained in a large fMRI database of PDS (with baseline controls) and used to inform clinical diagnosis or prognosis for patients with neuropathological disorders[53]. A recent study using a neurobiologically informed whole-brain computational model demonstrated that a change of excitatory-inhibitory balance in favor of inhibition can reproduce brain dynamics that characterize both PDS and UWS[54]. Hence, it is conceivable that an overall increase in neuronal inhibition may be mechanistically relevant to the degradation of cortical gradients, a proposition that warrants future investigation.

Our findings also have relevance for the systems-level neuronal mechanisms of psychiatric disorders, particularly SCHZ. We found that SCHZ can be characterized by a degradation of the cortical gradient from visual to somatomotor areas (Gradient-2). At the network level, SCHZ was associated with SMN-dominant changes in the gradient space, including shortened network distance of SMN-VIS, VAN-SMN, and FPN-SMN. These changes suggest a disruption of the normal functional segregation between the sensorimotor and other systems involved in multisensory integration and cognitive inferences. This could allow sensory information to "leak" erroneously into other systems due to porous cognitive borders, resulting in aberrant information transfer in SCHZ[55]. Finally, neither bipolar disorder nor ADHD showed a change in any gradient range (Fig. 3), which is to be expected

since, compared with the other states studied here, neither state nor content of consciousness is as dramatically affected in these psychiatric diseases.

There are multiple limitations of our study. First, we focused only on the first three cortical gradients that have been well documented in the literature[12,27–31]. The optimal number of cortical gradients needed to characterize the neurofunctional dimensions of consciousness remains to be determined. Second, we related the dimensions of consciousness to cortical gradients alone. Functional gradients in subcortical regions were not examined due to methodological constraints. We acknowledge that, besides the cerebral cortex, subcortical regions including the thalamus may play a significant role in shaping the brain's functional organization[56,57] relevant to conscious processing[58]. Although gradient analysis can be applied to the subcortex[59], subcortical gradients should preferably be derived from data with high spatial resolution, such as those obtained by ultrahigh field strength 7T-fMRI. However, this limitation might be mitigated because a recent study demonstrated that thalamocortical connectivity recapitulates large-scale cortical gradients[60]. Namely, distinct populations of cells within the thalamus (e.g., granular-projecting "core" cells vs. supragranular-projecting "matrix" cells) interact with cortical regions organized into distinct gradient zones, suggesting that the cortical gradients already incorporate the influence of thalamocortical interactions. Nevertheless, the extent to which cortical gradients are shaped by thalamic vs. corticocortical activity remains to be determined. Third, we had a limited number of patients with UWS ($n = 7$) suitable for cortical gradient analysis, because 6 out of 13 from our original sample had minor cortical distortions resulting in missing values for at least one of 400 cortical areas from the pre-defined brain parcellation scheme[26]. Additional methods may need to be developed for parcellation in brain-injured patients.

We conclude that cortical gradients can represent neurofunctional dimensions of consciousness, which advances the field beyond conceptual multidimensional frameworks. We present evidence that the cortical gradients of unimodal to transmodal, visual to somatomotor, and visual/default-mode to multiple-demand represent three neurofunctional dimensions of consciousness, i.e., awareness, sensory organization, and cortical arousability. Furthermore, disruptions of consciousness are associated with a degradation of one or more of the major cortical gradients, which vary depending on the state. Our work thus provides a unifying framework for a multidimensional model of human consciousness in both health and disease.

## Methods

### Dataset-1: propofol deep sedation

The dataset has been previously published using analyses different from those applied here[21,61]. The University of Michigan Institutional Review Board (IRB) approved the experimental protocol. All methods were performed in accordance with the relevant guidelines and regulations. Twenty-six healthy participants were recruited (male/female: 13/13; age: 19–34 years; right-handed). Informed consent was obtained from all participants, who were compensated for participation after the experiment. All participants were classified as American Society of Anesthesiologists physical status 1.

Before the study, participants fasted for eight hours. An attending anesthesiologist performed a preoperative assessment and physical examination on the day of the experiment. Throughout the experiment, two fully trained anesthesiologists were physically present and they continuously monitored spontaneous respiration, end-tidal $CO_2$, heart rate, pulse oximetry, and electrocardiogram. An automatic monitor compatible with MR was used to measure noninvasive arterial pressure. Following subcutaneous injection of lidocaine (0.5 ml of 1%) as a local anesthetic, an intravenous cannula was placed. Participants were provided with supplemental oxygen (2 L/min via nasal cannula).

Our reference drug was propofol, the most commonly used sedative-hypnotic in fMRI studies on the effects of anesthesia in humans. Propofol exerts minimal effects on cerebral hemodynamics and can be carefully titrated. Propofol modulates widespread targets throughout the brain by enhancing GABA-A receptor-mediated inhibition, which suppresses neuronal activity[36]. Propofol administration was achieved by target-controlled IV bolus and constant rate infusion. The bolus dose, infusion rate, and infusion duration for each target effect site concentration were predetermined based on a Marsh model implemented in STANPUMP (http://opentci.org/code/stanpump). To reach the final target, dosing (bolus plus infusion) was incremented every five minutes. Using incremental dosing (0.4 g/ml), the anesthetic level was titrated until no behavioral response was detected. There were 14 participants whose initial effect site concentration was 0.4 μg/ml and 12 participants whose initial effect site concentration was 1.0 μg/ml. The final effect site concentration was 2.4 μg/ml in six participants[62], and one increment above that resulted in loss of behavioral responsiveness in 20 participants. The final target was maintained for 21.6 min on average (±SD = 10.2 min). The infusion was then stopped to allow spontaneous emergence.

Behavioral responsiveness was assessed by hand-squeezing a rubber ball, which defined the periods of PDS (i.e., loss of behavioral responsiveness). Behavioral responses were measured in mmHg of air pressure using BIOPAC (https://www.biopac.com) MP160 system with AcqKnowledge software (V5.0). A total of 60 motor response trials were conducted during the entire scan, with an interval of ~90 s across trials. The trial was initiated by the spoken word "action", after which participants were instructed to grip the rubber ball once. The participants were instructed to perform mental imagery tasks in between motor response trials (playing tennis, spatial navigation, and squeeze imagery). In a pseudo-randomized (Latin square) block design, 15-s periods of tennis (and navigation) imagery, 10-s periods of squeeze imagery followed by a hand squeeze within 5-s, were alternated with 15-s of rest. A total of 180 rest-imagery cycles (60 cycles per condition) were completed throughout the scan. Mental imagery trials were cued with the spoken words "tennis imagery," "navigation imagery," and "squeeze imagery," and the rest period was cued with the word "relax." Verbal instructions were programmed using E-Prime 3.0 (Psychology Software Tools, Pittsburgh, PA) and delivered via an MRI-compatible audiovisual stimulus presentation system. More details about experimental design can be found in our previous publication[62].

Data were acquired using a 3 T Philips scanner with a standard 32-channel transmit/receive head coil at Michigan Medicine, University of Michigan. For high spatial resolution of anatomical images, T1 weighted spoiled gradient recalled echo images were acquired with the following parameters: 170 sagittal slices, 1.0 mm thickness (no gap), TR = 8.1 s, TE = 3.7 ms, flip angle = 8°, FOV = 240 mm, image matrix 256 × 256. A gradient-echo EPI pulse sequence was used to acquire functional images of the whole brain with the following parameters: 28 slices, TR/TE = 800/25 ms by multiband acquisition, MB factor = 4, slice thickness = 4 mm, in-plane resolution = 3.4 × 3.4 mm; FOV = 220 mm, flip angle = 76°, image matrix: 64 × 64. Six participants were scanned with slightly different parameters before MRI hardware upgradation (21 slices, TR/TE = 800/25 ms, MB factor = 3, slice thickness = 6 mm). In the scanner, participants were asked to remain awake with their eyes closed. Verbal instructions were delivered through earphones. Four fMRI runs were conducted, including 15-min conscious baseline, during (30-min) and after (30-min) propofol infusion, and another 15-min recovery baseline. The fMRI data acquired during conscious baseline and PDS were used in the present study.

## Dataset-2: propofol general anesthesia

The dataset has been previously published using analyses different from those applied here[15,63,64]. The study was approved by the IRB of Huashan Hospital, Fudan University. Twenty-six participants were recruited (male/female: 12/14; age: 27–64 years; right-handed). Informed consent was obtained from all participants, who were compensated for participation after the experiment. The participants were classified as American Society of Anesthesiologists physical status I or II, who were undergoing an elective trans-sphenoidal approach for pituitary microadenoma resection. Radiological examinations and plasma endocrine parameters were used to diagnose the pituitary microadenomas based on their size (<10 mm in diameter without growing out of the sellar region). No participants had a history of brain dysfunction, vital organ dysfunction, or neuropsychiatric medication use. There were no contraindications to MRI study, such as metal implants or vascular clips. Three participants were excluded from the study due to excessive movements, resulting in 23 participants.

Prior to the study, participants fasted for at least 8 h from solid food and 2 h from liquids. During the fMRI experiment, blood pressure, electrocardiography, pulse oximetry, and partial pressure of carbon dioxide were continuously monitored. Propofol light sedation (17 of 23 participants) and general anesthesia (23 participants) were administered through an intravenous catheter placed into a vein of the right hand or forearm. Propofol was administered using a target-controlled infusion (TCI) pump to obtain constant effect site concentration as estimated by Marsh model. Under general anesthesia, remifentanil (1.0 µg/kg) and succinylcholine (1.5 mg/kg) were administered for ease of endotracheal intubation. Starting at 1.0 mg/ml, TCI concentrations were increased in 0.1 mg/ml steps until reaching the appropriate target effect site concentration. The distribution of propofol between compartments was equilibrated over a 5-min period. During light sedation and general anesthesia, propofol concentrations were maintained at 1.3 µg/ml and 4.0 µg/ml, respectively.

The Ramsay scale was used to assess behavioral responsiveness. A participant was considered fully conscious (Ramsay 1–2) if they responded clearly and strongly to a verbal command ("strongly squeeze my hand!"), or considered mildly sedated if the response was clear but slow (Ramsay 3–4), or considered deeply sedated or anesthetized if there was no response (Ramsay 5–6). The Ramsay scale verbal command was repeated twice for each assessment. During conscious resting state and light sedation, participants breathed spontaneously with supplemental oxygen via nasal cannula. During general anesthesia, participants were ventilated with intermittent positive pressure ventilation, maintaining a tidal volume of 8–10 ml/kg and a respiratory rate of 10–12 beats per minute. Two certified anesthesiologists were present throughout the study. The participants wore earplugs and headphones during the fMRI scans.

Three 8-min fMRI scans were performed in conscious baseline, propofol light sedation, and PGA. Participants' heads were fixed in the scan frame and cushioned with spongy cushions to reduce head motion. During scanning, participants were asked to relax, assume a comfortable supine position, and close their eyes (an eye patch was applied). While undergoing resting-state scanning, they were instructed not to pay attention to anything in particular. A Siemens 3 T scanner (Siemens MAGNETOM, Germany) with a standard 8-channel head coil was used to acquire gradient-echo EPI images of the whole brain with the following parameters: 33 slices, TR/TE = 2000/30 ms, slice thickness=5 mm, FOV = 210 mm, flip angle=90°, image matrix: 64×64. High-resolution anatomical images were also acquired. The fMRI data acquired during conscious baseline and PGA were used in the present study.

## Dataset-3: ketamine anesthesia

The dataset has been previously published using analyses different from those applied here[15]. The study was approved by the IRB of Huashan Hospital, Fudan University. Twelve participants were recruited (male/female: 7/5; age: 32–66 years; right-handed). Informed consent was obtained from all participants, who were compensated for participation after the experiment. The participants were classified as American Society of Anesthesiologists physical status I or II, who were undergoing an elective trans-sphenoidal approach for pituitary microadenoma resection. The anesthesia procedure and fMRI scanning parameters were the same as those of Dataset-2.

An intravenous catheter was inserted into the left forearm vein to administer ketamine. For the entire experiment, fMRI scanning continued for about one hour, ranging between 44 min and 62 min. A 10-min conscious baseline was first acquired, except for two participants in which baseline condition was for 6 and 11 min. After that, 0.05 mg/kg/min of ketamine was infused for 10 min (0.5 mg/kg in total), followed by 0.1 mg/kg/min for another 10 min (1.0 mg/kg in total; except for two participants who received 0.1 mg/kg/min infusion for 10 min). Afterwards, the infusion of ketamine was discontinued, and participants spontaneously regained responsiveness.

Behavioral responsiveness was assessed during the fMRI scan. The verbal instruction "press the button" was programmed to play every 30 s using E-Prime 2.0 (Psychology Software Tools, Pittsburgh, PA) and delivered via an MRI-compatible audiovisual stimulus presentation system. Subjects' comfort was ensured by adjusting the volume of the headphones. Participants were instructed to press a button with their right index finger. The period of loss of behavioral responsiveness, defined as KA, was determined by comparing the timing of verbal instruction and actual responsiveness during and after ketamine infusion. The fMRI data length of KA was 18.2 ± 7.6 min across participants. The fMRI data acquired during conscious baseline and KA were used in the present study.

## Dataset-4: patients with neuropathological disorders

The dataset has been previously published using analyses different from those applied here[15]. The study was approved by the IRB of Huashan Hospital, Fudan University. Informed consent was obtained from the patients' legal representatives and from the healthy participants. They were compensated for participation after the experiment. The dataset included 21 patients with disorders of consciousness (male/female: 18/3; age: 8–78 years). The patients were assessed by the Coma Recovery Scale-Revised (CRS-R) on the day of fMRI scanning. Of those assessed, 13 patients were diagnosed as UWS, and 8 were diagnosed a minimally conscious state. The fMRI data acquired in healthy controls (n = 16; male/female: 8/8; age: 23–65 years) and seven patients with UWS were used in the present study. Of note, 6 out of 13 patients with UWS had to be excluded due to cortical distortions resulting in missing values in at least one of 400 cortical areas from the pre-defined brain parcellation scheme[26]. There was no history of neurological or psychiatric disorders or medication use among healthy controls.

A Siemens 3 T scanner (Siemens MAGNETOM, Germany) with a standard 8-channel head coil was used to acquire gradient-echo EPI images of the whole brain with the following parameters: 33 slices, TR/TE = 2000/35 ms, slice thickness = 4 mm, field of view = 256 mm, flip angle = 90°, image matrix: 64 × 64. Two hundred EPI volumes of resting-state fMRI as well as high-resolution anatomical images were acquired.

## Dataset-5: patients with psychiatric disorders

The dataset was obtained from the OpenNeuro database, which is shared by the UCLA Consortium from Neuropsychiatric Phenomics[65]. The study was approved by the Institutional Review Boards at UCLA and the Los Angeles County Department of Mental Health. Informed consent was obtained from all participants, who were compensated for participation after the experiment[65]. The original dataset included 272 participants with ages 21–50 years encompassing healthy individuals (n = 130) and individuals with psychiatric disorders including SCHZ

($n = 50$), BD ($n = 49$), and ADHD ($n = 43$). Detailed population characteristics can be found in ref. 65. Data were excluded if there were no T-1 images or resting-state data, if the overall head motion range exceeded 3 mm, or if the degree of freedom was insufficient after motion scrubbing and band-pass filtering[15]. Accordingly, 116 healthy individuals, 44 SCHZ, 49 BD, and 39 ADHD were included in the present study (248 in total).

## fMRI data preprocessing

Preprocessing steps were implemented in AFNI (linux_ubuntu_16_64; http://afni.nimh.nih.gov/). (1) Discarding the first two fMRI frames of each scan; (2) Slice timing correction; (3) Rigid head motion correction/realignment. Frame-wise displacement (FD) of head motion was defined as frame-wise Euclidean Norm of the six-dimension motion derivatives. If a frame's derivative value has a Euclidean Norm exceeds FD of 0.4 mm[66], this frame and its previous frame were excluded; (4) Coregistration with T1 anatomical images; (5) Spatial normalization into Talaraich stereotactic space; (6) Using AFNI's function 3dTproject, the time-censored data were band-pass filtered to 0.01–0.1 Hz. Various undesired components, including linear and nonlinear drift, time series of head motion and its temporal derivative, and mean time series from the white matter and cerebrospinal fluid, were removed via linear regression; (7) Spatial smoothing with 6 mm full-width at half-maximum isotropic Gaussian kernel; (8) Each voxel's time-course was normalized to zero mean and unit variance. GSR was applied for our main analysis, which presumably minimize unwanted global confounds such as low-frequency respiratory volume and cardiac rate[67]. The GSR procedure does not seem to affect the results derived from cortical gradient analysis[29]. Nevertheless, to evaluate the robustness of our results against different processing schemes, we also performed control analyses without applying the GSR procedure.

## Cortical gradient analysis

Based on a well-established brain parcellation scheme[26], the fMRI time courses were extracted from 400 pre-defined cortical areas[68] after preprocessing. For each participant and condition, a $400 \times 400$ connectivity matrix was constructed using Pearson correlation. The group-average (per condition) was calculated by averaging these individual connectivity matrices. Cortical gradients were computed using the BrainSpace toolbox (https://brainspace.readthedocs.io/en/latest/)[31] as implemented in MATLAB R2017b. As in previous work[12,29,30], we z-transformed and thresholded the connectivity matrix at the sparsity of 90%, i.e., leaving only the top 10% of weighted connections per row, and calculated a normalized cosine angle affinity matrix that captures the similarity of connectivity profiles between cortical areas. Using a diffusion map embedding algorithm, we identified gradient components, which estimated the low-dimensional embedding from the high-dimensional connectivity matrix. The algorithm is controlled by parameters $\alpha$ and $t$, where $\alpha$ controls the influence of the density of sampling points on the manifold ($\alpha = 0$, maximal influence; $\alpha = 1$, no influence) and $t$ controls the scale of eigenvalues of the diffusion operator. According to previous recommendations[12,27,29–31], we fixed $\alpha$ at 0.5 and $t$ at 0, preserving global relations between data points in the embedded space. Here $t = 0$ indicates that the diffusion time is derived from an automated estimation using a damped regularization process[12,31]. Based on Procrustes rotation, group-level gradient solutions were aligned to a subsample of the HCP dataset ($n = 217$) available in Brainspace toolbox[31]. Specifically, given a source **S** and a target **T** representation, Procrustes analysis seeks an orthogonal linear transformation $\psi$ to align the source to the target, such that $\psi$**S** and **T** are superimposed (mathematical details can be found in ref. 69). The reason for performing such transformation was that the gradients computed separately from different individuals may not be directly comparable due to different eigenvector orderings in case of eigenvalue multiplicity and sign ambiguity of the eigenvectors. As a result of

the alignment step, gradient estimation is more stable and solutions are more comparable to those from existing literature[31]. We then calculated individual-level gradients for each condition using identical parameters. In order to depict the cortical gradient organization at the network level, the gradient eigenvector loading values were extracted from seven pre-defined functional networks[26]. Control analyses were performed by varying the sparsity from 0% to 90% (by 10% increments) and varying the parameter $\alpha$ from 0 to 1 (by 0.1 increments).

## Distance metrics in the cortical gradient space

We calculated a set of measures to quantify the range of each gradient, global dispersion, network eccentricity, and distance between functional networks in a three-dimensional gradient space. These measures were calculated for each participant within the individualized, aligned gradient space: (1) Numerical range of each gradient was calculated as the distance from the minimum to the maximum gradient eigenvector values, indicating segregation (i.e., different connectivity profile) of the gradient extremes; (2) Global dispersion was quantified as sum squared Euclidean distance of all regions to the global centroid within the 3D gradient space. A small dispersion value indicates that the functional connectivity profiles across regions have a low differentiation along three gradients; (3) Network eccentricity was calculated as the squared Euclidean distance between a given network centroid and the global centroid; (4) Between-network distance was calculated as the squared Euclidean distance between-network centroids. These multidimensional distance metrics reflect the similarity of connectivity profiles between cortical areas across multiple axes of differentiation[27].

## Tracking large-scale co-activation patterns

We calculated the spatial similarity between the signal intensity of each fMRI volume and eight pre-defined centroids of co-activation patterns (CAPs)[15]. The CAPs included the DMN, DAN, FPN, SMN, VIS, VAN, and global network of activation and deactivation (GN+ and GN−). We assigned each fMRI volume to a particular CAP based on its maximal similarity to the CAP centroids. Accordingly, this produced a time series of discrete CAP labels. The individual-level occurrence rate of each CAP was quantified by the ratio of the number of volumes that appeared versus the total number of volumes per state/condition.

## Statistics and reproducibility

For an fMRI study of cognitive function, Desmond and Glover[70] reported that about 25 participants are necessary to achieve 80% power for a 0.5% increase in activity. We conducted power analysis based on our previous study with graded sedation of propofol[71]. The average Cohen's d as a measure of the effect size was 0.56. For 80% power and $\alpha = 0.05$, 24 human participants were needed. The obtained participants in Dataset-1 ($n = 26$) and Dataset-2 ($n = 23$) were above or very close to the suggested optimal group size. The number of participants/patients in Dataset-3 ($n = 12$) and Dataset-4 ($n = 16, 7$) were limited. In addition, we evaluated the specificity of our results in an independent cohort of 248 participants consisting of healthy control participants and patients with psychiatric disorders. The experiments were not randomized because within-subject design was used in Dataset-1, Dataset-2, and Dataset-3. Dataset-4, and Dataset-5 were collected from independent research sites with different research protocols. The investigators were not blinded to allocation during experiments and outcome assessment.

Using Bayesian statistics as implemented in JASP (v0.16.3; https://jasp-stats.org/) with default effect size priors, Cauchy scale 0.707[32], the distance metrics (i.e., gradient range, global dispersion, network eccentricity, network distance) were compared between baseline condition and a given depressed state of consciousness or psychiatric diagnosis. Bayesian Paired Samples $T$ tests (two-tailed) were performed for PDS, PGA, and KA against their own baseline conditions. Bayesian Independent Samples $T$ tests (two-tailed) were performed for UWS, SCHZ, bipolar disorder, and ADHD against their own healthy

control groups. Results were reported using two-tailed Bayes factor $BF_{10}$ that represents $p(data|H_1$: there is effect$)/p(data|H_0$: there is no effect). Posterior distribution with a measure of median and 95% credible interval (CI) were reported. To minimize potential false inferences due to the numerous statistical tests, we excluded anecdotal and moderate evidence ($1 < BF_{10} < 10$), while collected strong, very strong, and decisive evidence ($BF_{10} > 10$) for $H_1$ (there is effect) or against ($H_0$: there is no effect) either hypothesis[32].

Classical Student's $T$ tests (paired or independent samples) were also performed for calculating $t$ and $p$ values. Using the Benjamini–Hochberg procedure, $p$ values were false discovery rate-corrected (FDR-corrected) for multiple comparisons for each dataset and thresholded at $\alpha = 0.05$. In addition, assumption checks were performed with Shapiro–Wilk test for normality and Levene's test for homogeneity of variances. While the majority of comparisons satisfied both data normality and equal variances, there were some cases that violated one of the two. We, therefore, performed non-parametric tests for those cases. Specifically, Bayesian Wilcoxon signed-rank tests were applied to paired samples, and Bayesian Mann–Whitney $U$ tests were applied to unpaired samples. The detailed statistics are reported in Supplementary Data 1 and Supplementary Data 2. The results based on non-parametric tests were qualitatively similar to our main results with parametric tests.

Bayesian Kendall Correlations were performed between-network distance (e.g., DMN-VAN, DMN-DAN, and VAN-DAN) and occurrence rates (e.g., VAN+, DAN+, and DMN+) across all subjects and conditions. Results were reported with Kendall's tau-b (τb) and Bayes factor $BF_{10}$ that represents $p(data|H_1$: the population correlation takes its value between $−1$ and $1)/p(data|H_0$: the correlation between pairs of variables equal 0). We applied Bayesian Kendall Correlations instead of Bayesian Pearson Correlations, because the Shapiro–Wilk Test indicates a violation of normality for most of the correlation pairs ($p < 0.05$). In addition, classical Spearman rank correlations were performed for calculating $p$ values with FDR-corrected at $\alpha = 0.05$.

One of the key findings, i.e., degradation of Gradient-1 induced by propofol anesthetics, was first found in Dataset-1 and reproduced independently in Dataset-2.

### Reporting summary
Further information on research design is available in the Nature Portfolio Reporting Summary linked to this article.

## Data availability
The re-used datasets in this study have been published[15,21], which are publicly available from Zenodo repository (https://doi.org/10.5281/zenodo.6955280). The psychiatric dataset re-used in this study is publicly available from OpenfMRI (https://legacy.openfmri.org/dataset/ds000030/). Source data to plot the figures are provided with this paper. Source data are provided with this paper.

## Code availability
Custom code is publicly available from Zenodo repository (https://doi.org/10.5281/zenodo.6955280).

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

## Acknowledgements

This study was supported by a grant from the National Institute of General Medical Sciences of the National Institutes of Health under Award R01-GM103894 (to A.G.H.). The content is solely the responsibility of the authors and does not necessarily represent the official views of the National Institutes of Health. Z.H. was supported by the University of Michigan Anesthesiology Post-Doctoral Research Training Program (NIGMS; 5T32GM103730-09). The authors thank Drs. Georg Northoff and Jun Zhang who shared Dataset-2 and Dateset-3.

## Author contributions

Z.H. conducted the experiment and collected Dataset-1 and Dataset-4. Z.H. analyzed all datasets and prepared the figures. Z.H. and A.G.H. designed the study. Z.H., G.A.M., and A.G.H. interpreted the data and wrote the manuscript.

## Competing interests

The authors declare no competing interests.
