## [Peer Review File · Nature Communications]

Functional geometry of the cortex encodes dimensions of consciousnessEditorial Note: This manuscript has been previously reviewed at another journal that is not operating a transparent peer review scheme. This document only contains reviewer comments and rebuttal letters for versions considered at *Nature Communications*.

Parts of this Peer Review File have been redacted as indicated to remove third-party material where no permission to publish could be obtained.

REVIEWER COMMENTS

Reviewer #1 (Remarks to the Author):

I thank the authors for their careful review of their work.

Please find below some comments and suggestions that I hope may help the manuscript:

1. Possibly this work is interesting to consider in the light of the current paper <https://www.ncbi.nlm.nih.gov/pmc/articles/PMC7907463/>
2. Regarding the remark on the sub cortex, isn't it unclear to what extent the similarity between thalamic-cortical and cortico-cortical connectivity may reflect that cortico-cortical interactions are shaped by thalamic-cortical connectivity?

best wishes

Reviewer #2 (Remarks to the Author):

Thank for your response. All my concerns and questions were adequately answered.

Reviewer #3 (Remarks to the Author):

The authors examine the relationship between states of consciousness and large-scale gradient organization of functional connectivity. The study uses several unique datasets and analyses. The authors have conducted a detailed analysis using cortical gradients. In the revised version, the authors have done a good job of addressing previous concerns. However, major concerns remain.

This study extends the authors' important previous work "Temporal circuit of macroscale dynamic brain activity supports human consciousness" showing that isolation of the default mode and dorsal attention networks from the temporal circuit is associated with unresponsiveness of diverse etiologies under different anesthetics (Huang et al. *Sci Adv*, 2020). Here the authors use the same data and examine it with a different analysis technique. While the authors' approach is innovative, the rationale for the study is not clearly spelt out. What were the authors trying to achieve beyond their previous study which also focused on the DMN, DAN and VAN? Why would the analysis of cortical gradients present an advance? No links are provided to the previous study as a result why the newer results are an advance is not clear.

A number of complex analyses are presented but are not tied in together in cohesive manner. The study moves back and forth between gradient and network analyses in a manner that is confusing and not hypothesis driven. A clearer articulation of the hypotheses and goals is needed.

While the organization of principal gradient 1 is clear, and consistent with Margulies et al 2016, the case for other gradients is less clear in the context of various anesthetics used in the study. The issue of dosage versus anesthetic-type remains.

Crucially, the main result "Macroscale cortical gradients define multidimensional states of consciousness" is based on the analysis reported in Figure 5. While the distance measure is based on cortical gradients, the analysis is mainly based on DMN, DAN and VAN interactions reported in

the previous study. The new findings appear incremental.

Overall, this an interesting study using a unique set of data. However, the novelty of the findings and advances over the prior study are less clear. The authors need to articulate the new research questions and hypotheses better, rather than merely apply cortical gradient models to their data. Doing so will make potential advances clearer to the reader.

Finally, the data should be shared more transparently via openfmri, as the authors themselves have availed of this resource.

Reviewer #1 (Remarks to the Author):

I thank the authors for their careful review of their work. Please find below some comments and suggestions that I hope may help the manuscript:

1. Possibly this work is interesting to consider in the light of the current paper <https://www.ncbi.nlm.nih.gov/pmc/articles/PMC7907463/>

Response:

This is a very interesting review article, in which the discussion about gradient-3 (Figure 5 in Smallwood et al. 2021) is of particular relevance. We have cited this paper in Discussion of our revised manuscript (new edits are highlighted in green font):

“We postulate that Gradient-3 is associated with arousal. The conventional concept of arousal relates to the ascending reticular activating system (ARAS), which includes the brainstem, medial thalamus, and basal forebrain. Hence, Gradient-3 is best interpreted as cortical arousal, which could be a result of ARAS activation. Further study is required to identify a covariation between Gradient-3 range and regional activation along the ARAS. Nevertheless, the neurofunctional role of Gradient-3 remains to be determined. **First, this gradient was previously identified as paralimbic to multiple-demand²⁸ or task-negative to task-positive⁴² functional axis, and was suggested to be relevant for cognitive processes involving vigilance and working memory²⁸, and the effect of task demands (off-task vs. on-task) on cognition⁴².**” (line 28-31, page 10; line 1-5, page 11)

42. Smallwood, J. et al. The neural correlates of ongoing conscious thought. *iScience* 24, 102132 (2021).

2. Regarding the remark on the sub cortex, isn't it unclear to what extent the similarity between thalamic-cortical and cortico-cortical connectivity may reflect that portico-cortical interactions are shaped by thalamic-cortical connectivity?

Response:

We fully agree with reviewer’s comment. We have revised related statements in the limitations:

“Second, we related the dimensions of consciousness to cortical gradients alone. Functional gradients in subcortical regions were not examined due to methodological constraints. We acknowledge that, besides the cerebral cortex, subcortical regions including the thalamus may play a significant role in shaping the brain’s functional organization^{56,57} relevant to conscious processing⁵⁸. Although gradient analysis can be applied to study the subcortex⁵⁹, the subcortical gradients should preferably be calculated from data with high spatial resolution, such as those obtained by ultrahigh field strength 7T-fMRI. However, this limitation might be mitigated because a recent study demonstrated that thalamocortical connectivity recapitulates large-scale cortical gradients⁶⁰. Namely, distinct populations of cells within the thalamus (e.g., granular-projecting ‘core’ cells vs. supragranular-projecting ‘matrix’ cells) interact with cortical regions organized into distinct gradient zones, suggesting that the cortical gradients already incorporate the influence of thalamocortical interactions. Nevertheless, the extent to which cortical gradients are shaped by thalamic vs. corticocortical activity remains to be determined.” (line 5-17, page 15)

Two new references have been added to support the role of thalamus in shaping the brain’s functional organization, as follows:

56. Bell, P. T. & Shine, J. M. Subcortical contributions to large-scale network communication. *Neurosci. Biobehav. Rev.* 71, 313–322 (2016).

57. Shine, J. M. et al. The low-dimensional neural architecture of cognitive complexity is related to activity in medial thalamic nuclei. *Neuron* 104, 849-855.e3 (2019).

Reviewer #2 (Remarks to the Author):

Thank for your response. All my concerns and questions were adequately answered.

Response:

We appreciate the positive comments.

Reviewer #3 (Remarks to the Author):

The authors examine the relationship between states of consciousness and large-scale gradient organization of functional connectivity. The study uses several unique datasets and analyses. The authors have conducted a detailed analysis using cortical gradients. In the revised version, the authors have done a good job of addressing previous concerns. However, major concerns remain.

- 1. This study extends the authors' important previous work "Temporal circuit of macroscale dynamic brain activity supports human consciousness" showing that isolation of the default mode and dorsal attention networks from the temporal circuit is associated with unresponsiveness of diverse etiologies under different anesthetics (Huang et al. Sci Adv, 2020). Here the authors use the same data and examine it with a different analysis technique. While the authors' approach is innovative, the rationale for the study is not clearly spelt out. What were the authors trying to achieve beyond their previous study which also focused on the DMN, DAN and VAN? Why would the analysis of cortical gradients present an advance? No links are provided to the previous study as a result why the newer results are an advance is not clear.***

Response:

We thank the review for this important remark, which has motivated us to be clearer in the manuscript regarding the novelty of our work. Although we used previously published data, the current work is not an incremental extension of our previous work (Huang et al. Sci Adv, 2020). The current study is novel because, for the first time, we use the analysis of cortical gradients to identify neurofunctional substrates as the key dimensions of consciousness. This is an interval advance for the field because existing multidimensional frameworks for consciousness are conceptual rather than neurobiological. The possibility of cortical gradients as candidates for neurofunctional dimensions of consciousness was not hypothesized or addressed in our previous work or, to our knowledge, in any prior work. The dissection of the neurofunctional dimensions of consciousness through cortical gradient mapping thus represents the key advance (e.g., Figure 3). The network characterizations in the multidimensional gradient space (e.g., Figure 4 and Figure 5) and, pursuant to the Reviewer's point below, their correlations to dynamic brain states (e.g., Figure 6) were further investigations built upon cortical gradient mapping.

In the revised manuscript, we paid particular attention to more clearly articulating the novelty and hypotheses of our work, and have added a new figure (Figure 1) to highlight the relationship among the hypotheses (new edits are highlighted in green font):

“Recent advances in neuroimaging offer a new approach that can be applied to investigate the neurobiology of consciousness. Cortical gradients, which subsume multiple functions and functional networks along a continuum from the unimodal systems that underpin perception and action to the association cortices implicated in abstract cognitive functions¹²⁻¹⁴, are compelling candidates for the neurofunctional dimensions of consciousness. Our approach represents a conceptual departure from cognitive-behavioral definitions of the state of consciousness, such as the traditional two-dimensional representation (awareness vs. wakefulness)¹, or multiple cognitive representations⁵⁻⁸. The state of consciousness is defined here, independently of the cognitive-behavioral aspects, by a family of cortical gradients derived from the principled characterization of functional geometry.” (line 29-30, page 2; line 1-7, page 3)

“In this work, we analyze fMRI-derived cortical gradients to establish a multidimensional neurofunctional account of consciousness. We tested three hypotheses as follows (Fig. 1): (1) different neurofunctional dimensions of consciousness are represented by different cortical gradients, and the disruption of consciousness is associated with a degradation of one or more of the major cortical gradients, depending on the state; (2) cortical gradients construct a virtual multidimensional space where depressed or altered states of consciousness are associated with both common and state-specific alterations; (3) reconfigurations of brain network functional geometry are associated with a disruption of structured transitions of dynamic brain states, defined previously as a temporal circuit¹⁵. We test our hypotheses by comparing cortical gradients among large cohorts of healthy awake and anesthetized participants as well as patients with neuropathological and psychiatric diagnoses. We provide an empirical basis for a unifying neurofunctional framework in which (1) dimensions of consciousness are defined in neural terms, (2) these dimensions are characteristically altered depending on the state of consciousness, and (3) changes in the brain’s functional geometry also influence temporal dynamics.” (line 16-28, page 3)

[REDACTED]

Fig. 1 Schematic illustration of three working hypotheses.

- 2. A number of complex analyses are presented but are not tied in together in cohesive manner. The study moves back and forth between gradient and network analyses in a manner that is confusing and not hypothesis driven. A clearer articulation of the hypotheses and goals is needed.*

Response:

Thank you. This has been addressed in Point #1.

- 3. While the organization of principal gradient 1 is clear, and consistent with Margulies et al 2016, the case for other gradients is less clear in the context of various anesthetics used in the study. The issue of dosage versus anesthetic-type remains.*

Response:

Thank you for the comment. Margulies et al. (2016) primarily reported and discussed the principal gradient-1 in the main body of their research article. They also reported other gradients in their Supporting Information Fig. S1. The gradients derived from our data are comparable with those reported from Margulies et al., as well as other studies (e.g., Bethlehem et al., Neuroimage, 2020; Hong et al., Nat. Commun., 2019; Vos de Wael et al., Commun. Biol., 2020; Mckeown et al., Neuroimage, 2020; Cross et al., Neuroimage, 2021). See our Supplementary Fig. 1 for the topographic profiles of the cortical gradients across all conditions.

As for the issue of dosage versus anesthetic-type, both of them are factors of interest, instead of confounds, in our study.

We stated that:

“We demonstrated that the effects of these two drugs on cortical gradients were dissociable along two different dimensions, i.e., unimodal to transmodal (Gradient-1) and visual to somatomotor (Gradient-2). A plausible inference is that propofol suppresses awareness by collapsing a unimodal-transmodal hierarchical processing without changing sensory organization, whereas ketamine partially preserves inner subjective experience but distorts sensory experience by collapsing different sensory modalities. In addition, we observed dose-dependent effects of propofol on cortical gradients, where a high dose further collapsed the cortical gradient of visual/default-mode to multiple-demand areas (Gradient-3).” (line 18-25, page 13)

We also acknowledged that:

“Accordingly, there remains a possibility that ketamine anesthesia would also show dose-dependent effects on the cortical gradients. For example, a higher dose of ketamine might further collapse Gradient-1 or Gradient-3. Therefore, future studies linking the underlying molecular pathways and the cortical gradients with varied drug doses may be of relevance in testing these hypotheses.” (line 25-28, page 13)

4. Crucially, the main result “Macroscale cortical gradients define multidimensional states of consciousness” is based on the analysis reported in Figure 5. While the distance measure is

based on cortical gradients, the analysis is mainly based on DMN, DAN and VAN interactions reported in the previous study. The new findings appear incremental.

Response:

As discussed in Point #1, the main result supporting “Macroscale cortical gradients define multidimensional states of consciousness” is based on the analysis reported in **Figure 3** (previous Figure 2) instead of Figure 6 (previous Figure 5). The purpose of conducting the correlation analysis between network distance measures and occurrence rate of co-activation patterns in Figure 6 was to provide further insights on the relationship between the brain’s spatial and temporal characteristics, specifically, how a change in the brain’s functional geometry would affect the brain’s temporal dynamics. This had been addressed in the Discussion (line 18-30, page 12; line 1-8, page 13). The findings in Figure 6 are complementary to our main result of cortical gradients, and the main result could stand alone without the analysis reported in Figure 6.

5. Overall, this an interesting study using a unique set of data. However, the novelty of the findings and advances over the prior study are less clear. The authors need to articulate the new research questions and hypotheses better, rather than merely apply cortical gradient models to their data. Doing so will make potential advances clearer to the reader.

Response:

We thank the reviewer for the constructive comments. We believe that the new research questions and hypotheses have been clearly articulated in the revised manuscript, as addressed in Point #1 and Point #4.

6. Finally, the data should be shared more transparently via openfmri, as the authors themselves have availed of this resource.

Response:

This is a great suggestion. We are enthusiastic about sharing the data via OpenNeuro (formerly OpenfMRI). Some of the data are derived from our ongoing NIH-funded project (R01-GM103894).

We will be able to share the data with the scientific community after the project is completed, following additional data de-identification procedures per OpenNeuro requirement.

At this point, we have a statement in Data availability:

“Access to data by qualified investigators (i.e., affiliated with accredited academic and research institutions) are subject to scientific and ethical review and must comply with the National Institutes of Health (NIH) regulations. Completion of a material transfer agreement signed by an institutional official will be required in order to access the raw data.”

In addition, we have provided a zipped folder containing the data underlying all display items as ‘Source Data’ per journal request.

REVIEWERS' COMMENTS

Reviewer #1 (Remarks to the Author):

Thanks for the responses.

Reviewer #3 (Remarks to the Author):

The authors have addressed my concerns. The manuscript is improved as a result.